# Genome-Wide Identification of Seven in Absentia E3 Ubiquitin Ligase Gene Family and Expression Profiles in Response to Different Hormones in *Uncaria rhynchophylla*

**DOI:** 10.3390/ijms25147636

**Published:** 2024-07-11

**Authors:** Jinxu Lan, Conglong Lian, Yingying Shao, Suiqing Chen, Ying Lu, Lina Zhu, Detian Mu, Qi Tang

**Affiliations:** 1School of Pharmacy, Henan University of Chinese Medicine, Zhengzhou 450046, China; ianjx@163.com (J.L.); liancl00@163.com (C.L.); chsq@hactcm.edu.cn (S.C.); 2College of Horticulture, Hunan Agricultural University, Changsha 410128, China; syy2250519718@126.com (Y.S.); luying960522@163.com (Y.L.); zhulina_2021@163.com (L.Z.)

**Keywords:** *Uncaria rhynchophylla*, SINA, TIAs, bioinformatics, expression pattern, plant hormones

## Abstract

SINA (Seven in absentia) E3 ubiquitin ligases are a family of RING (really interesting new gene) E3 ubiquitin ligases, and they play a crucial role in regulating plant growth and development, hormone response, and abiotic and biotic stress. However, there is little research on the SINA gene family in *U. rhynchophylla*. In this study, a total of 10 *UrSINA* genes were identified from the *U. rhynchophylla* genome. The results of multiple sequence alignments and chromosomal locations show that 10 *UrSINA* genes were unevenly located on 22 chromosomes, and each UrSINA protein contained a SINA domain at the N-terminal and RING domains at the C-terminal. Synteny analysis showed that there are no tandem duplication gene pairs and there are four segmental gene pairs in *U. rhynchophylla*, contributing to the expansion of the gene family. Furthermore, almost all *UrSINA* genes contained the same gene structure, with three exons and two introns, and there were many cis-acting elements relating to plant hormones, light responses, and biotic and abiotic stress. The results of qRT-PCR show that most *UrSINA* genes were expressed in stems, with the least expression in roots; meanwhile, most *UrSINA* genes and key enzyme genes were responsive to ABA and MeJA hormones with overlapping but different expression patterns. Co-expression analysis showed that UrSINA1 might participate in the TIA pathway under ABA treatment, and UrSINA5 and UrSINA6 might participate in the TIA pathway under MeJA treatment. The mining of *UrSINA* genes in the *U. rhynchophylla* provided novel information for understanding the *SINA* gene and its function in plant secondary metabolites, growth, and development.

## 1. Introduction

Hypertension can cause many cardiovascular disease, thereby causing damage to many organs [1]. People are increasingly in need of an effective therapeutic schedule. *Uncaria rhynchophylla*, a traditional Chinese medicinal herb, has attracted widespread attention globally due to its bioactive ingredients, terpenoid indole alkaloids (TIAs), including rhynchophylline and isorhynchophylline, which have an excellent effect on hypertension [2]. Rhynchophylline and isorhynchophylline represent a class of TIAs that might be synthesized by strictosidine to act as the skeleton in *U. rhynchophylla* [3]. Strictosidine is formed through a coupling reaction between the final product of the iridoid pathway, secologanin, and the tryptamine of the shikimate pathway, catalyzed by strictosidine synthase (STR) [4]. Because the biosynthesis pathway of TIAs is complicated in *U. rhynchophylla*, further research is needed to elucidate it (Figure 1). In the natural environment, plants frequently encounter various biotic and abiotic stresses that hinder their survival. To adapt to these challenges, plants have developed diverse mechanisms to regulate their growth and development [5,6]. Plant growth and development is a highly complex process controlled by multiple hormones coordinating effects, including abscisic acid (ABA) [7]. ABA, as a plant hormone, plays an important role in signal transduction, morphogenesis, and especially secondary metabolism [8,9,10]. In *Catharanthus roseus*, the treatment of ABA in suspension cells increased catharanthine accumulation and up-regulated the expression of *CrSTR* and *CrTDC*. Treating suspension cells of *Taxus chinensis* with ABA can up-regulate the expression of related genes in the taxol biosynthesis pathway and *TcMYB29a*, and it can finally increase the content of the taxol. Moreover, ubiquitination-mediated protein degradation is an important process in ABA signal transduction [11]. Transcription factors, post-transcriptional modifications, and secondary metabolism play important roles in abiotic stress responses by regulating the expression of stress-responsive genes [12,13]. However, the modification and degradation mechanisms of the ubiquitination of functional proteins that regulate the TIA pathway in *U. rhynchophylla* are still unknown.

Ubiquitin, a small protein composed of 76 amino acids, plays a crucial role in the process of ubiquitination. The process of ubiquitination is catalyzed by the E1 ubiquitin-activating enzyme, E2 ubiquitin-binding enzyme, and E3 ubiquitin ligase, and proteins that undergo polyubiquitination are subsequently degraded by 26S proteasomes [14,15]. E3 ubiquitin ligase serves as a key enzyme in determining substrate specificity, facilitating the binding of ubiquitin molecules to the lysine residues of target proteins [16,17]. To date, hundreds of E3 ubiquitin ligases have been identified in plants, which can be mainly divided into four types, RING (really interesting new gene), U-box, HECT (Homology to E6-Associated Carboxy-Terminus), and Cullin-RING ligases (CRLs), according to their different reaction mechanisms [18]. Among them, the RING-type E3 ubiquitin ligase is present at the highest quantity and plays an important role in regulating seed germination and hormone response processes [19,20]. SINA (Seven in absentia) E3 ligases belong to the monomer RING-type E3 subfamily. This subfamily is characterized by a RING domain that binds to E2 at the N-terminus and a typical SINA domain that binds to substrates at the C-terminus [21]. SINA proteins were first identified in fruit flies. As on now, SINA proteins have been identified in many plants, such as *Arabidopsis thaliana*, *Oryza sativa*, *Actinidia chinensis*, and apple [22,23,24].

Ubiquitination plays a critical role in regulating ABA signaling by managing the stability and activity of proteins, along with the associated endomembrane transport. This regulation impacts the essential components of abscisic acid synthesis and signal transduction, ultimately influencing the plant’s response to ABA [25]. The main obstacle in comprehending ABA-signaling mechanisms lies in the identification of ABA receptors. Recent research has shown that ABA receptors and ABA-signaling proteins undergo ubiquitination, suggesting that ubiquitination plays a crucial role in the ABA-signaling pathway. In apple, almost all *MdSINA* genes are significantly changed by ABA treatment, and MdSINA2 has been proven to be an ABA-hypersensitive protein by its transgenic calli and heterologous expression in *Arabidopsis* [24]. In addition, E3 ubiquitin ligase MdSINA3 delays leaf senescence by targeting the degradation of MdBBX37, and MdBBX37 interacts with the ABA-signaling regulatory protein MdABI5, participating in ABA-mediated leaf senescence [26]. Furthermore, SINA proteins also play an important role in plant growth and development, defense response, and secondary metabolism [27,28,29]. A recent study demonstrated that the overexpression of *SlSINA2* and *SlSINA5* genes in tomatoes can result in light-green leaves and alterations in flower structure [27]. Additionally, in bread wheat, the E3 ligase TaSINA has been shown to enhance biomass and crop yield in the presence of heat stress [28]. In *Aquilaria sinensis*, a C3HC4-type E3 ubiquitin ligase, RING3, enhanced the content of sesquiterpene by degrading AsWRKY44 through ubiquitination [29].

Although SINA gene families have been identified in many plants, and their functions have been systematically studied, there is no report on the SINA E3 ubiquitin ligase family in *U. rhynchophylla*. In this research, 10 UrSINA family members were identified in *U. rhynchophylla*. Then, bioinformation analyses, including assessments of conserved motifs and gene structure, collinearity, cis-acting elements, expression profiles in different tissues, and their responses to different hormones, were implemented. The genome of U. rhynchophylla has provided an excellent platform to identify the SINA gene family and has laid the foundation for illuminating the mechanisms underlying the ubiquitination modification and degradation of functional proteins involved in regulating the synthesis of TIAs in *U*. *rhynchophylla.*

## 2. Results

### 2.1. Identification and Characterization of UrSINA Family Members

In this work, ten UrSINA genes were identified in the *U. rhynchophylla* genome, using Simple HMM search (E-value < 1 × 10^−5^) [30]. Then, Batch CD-search, Prosite, and the SMART online website were used to verify the integrity of the conserved domain (RING box and SINA domain), which was named UrSINA1-UrSINA10 in relation to its location on the chromosomes [31,32]. The basic physicochemical properties of UrSINA family members were as shown in Table 1. The number of amino acids varied from 206 aa (UrSINA6) to 506 aa (UrSINA7), corresponding to their molecular weights, which ranged between 23.63 kDa (UrSINA6) and 57.46 kDa (UrSINA7), with an average of 40.55 kDa. Among the ten UrSINA proteins, UrSINA2, UrSINA3, and UrSINA5 were basic proteins (pI > 7), and the remaining proteins were acidic proteins (pI < 7). Besides this, except for UrSINA4 and UrSINA7, which are stable proteins (instability index < 40), other SINA proteins are unstable proteins (instability index > 40). The vast majority of SINAs are predicted to be located in the mitochondrion and nucleus, and UrSINA1 and UrSIAN4 were only located in the nucleus.

### 2.2. Phylogenetic Tree Construction, Classification, and Multiple Sequence Alignment of UrSINA

To investigate the classification and phylogenetic relationships of UrSINAs, a phylogentic tree was constructed using SINA proteins from *A. thalina*, *O. sativa*, and *U. rhynchophylla*. According to the classification principles of AtSINA, the ten UrSINA proteins were divided into two groups, Group I and Group II (Figure 2). Three UrSINA proteins (UrSINA8, UrSINA9, and UrSINA10) were classified into Group I, and seven UrSINA proteins (UrSINA1, UrSINA2, UrSINA3, UrSINA4, UrSINA5, UrSINA6, and UrSINA7) were set as belonging to Group II. We found that ten UrSINA proteins had the typical RING domain at the N-terminal and a SINA domain at the C-terminal. Three UrSINA proteins, including UrSINA8, UrSINA9, and UrSINA10, showed an extended C-terminal compared with other UrSINAs, suggesting the distinct functions of UrSINA proteins (Figure 3).

### 2.3. Conserved Motifs and Gene Structure of UrSINA Family Members

To gain a deeper understanding of the characteristics of the UrSINA protein, MEME was used to analyze ten conserved motifs in ten UrSINA protein sequences (Figure 4). Motif 2, Motif 4, and Motif 7 were detected in all UrSINA proteins, while Motif 8 and Motif 10 were only present in UrSINA8, UrSINA9, and UrSINA10, which belonged to Group I. Furthermore, Motif 3, Motif 5, and Motif 9 were detected in almost all UrSINA proteins, except UrSINA1 and UrSINA6. Members from the same group, especially some closed members, contained similar conversed motifs. Moreover, in order to explore the gene structure of the UrSINA family member, intron/exon structure analysis was conducted. Almost all *UrSINA* genes have three exons and two introns, except *UrSINA1* and *UrSINA7*. *UrSINA1* genes have the fewest exons (*n* = 2) and introns (*n* = 1), and *UrSINA7* genes have nine exons and eight introns, respectively.

### 2.4. Chromosomal Localization and Intraspecific Synteny Analysis of UrSINA Gene Family

The chromosomal distribution of UrSINA family members was illustrated using TBtools (Figure 5a) [33]. Significantly, *UrSINA10* was only present on scaffold19 and not on the chromosomes. The remaining nine *UrSINA* genes were unevenly distributed on 22 chromosomes of *U. rhynchophylla*. The highest number of *SINA* genes was found on chromosome 8 (*UrSINA5* and *UrSINA6*). Chromosomes 2, 4, 5, 7, 14, 16, and 18 each had only one *UrSINA* gene, while no genes were distributed on the remaining 14 chromosomes.

Furthermore, segmental duplication events were analyzed using the Multiple Collinearity Scan toolkit (MCScanX) [34]. There were four pairs of segmental duplication between five chromosomes, which were Ur_chr4 (*UrSINA2*)/Ur_chr5 (*UrSINA3*), Ur_chr14 (*UrSINA7*)/Ur_chr16 (*UrSINA8*), Ur_chr14 (*UrSINA7*)/Ur_chr18 (*UrSINA9*), and Ur_chr16 (*UrSINA8*)/Ur_chr18 (*UrSINA9*). These results imply that segmental duplication events were the main factor affecting the expansion of the UrSINA gene family. In order to investigate the evolutionary relationships within the UrSINA gene family, interspecific collinearity analysis was conducted among SINA family members of *U. rhynchophylla* and three other species (*A.thaliana*, *O. sativa*, and *C. canephora*). The interspecific collinearity analysis revealed that *UrSINA* exhibited 11 collinear gene pairs with *C. canephora*, 10 collinear gene pairs with *A. thaliana*, and no collinear gene pairs with *O. sativa* (Figure 5c). These findings suggest a close relationship between *U. rhynchophylla* and *C. canephora*, which belong to the Rubiaceae family, while showing a more distant connection to monocotyledonous *O. sativa*. Moreover, *UrSINA1*, *UrSINA2*, *UrSINA3*, *UrSINA8*, and *UrSINA9* were all found to form collinear gene pairs with the *SINA* genes of *A. thaliana* and *C. canephora*, indicating that these genes might play an important role in the evolutionary process (Appendix A).

### 2.5. Analysis of Cis-Acting Elements in the Promoter of UrSINA Genes

Cis-regulatory elements are non-coding DNA sequences located in gene-promoter regions, which are important for gene expression and intimately participate in the regulation of various processes in plant growth and development [35]. PlantCare was used to predict potential cis-acting elements in the 2000 bp promoter region, exploring the functions of *UrSINAs*. Each *UrSINA* gene showed at least one cis-acting element in its promoter region (Figure 6a,b). Among these, 17 types of 119 cis-acting elements were identified in the promoter regions of 10 *UrSINA* genes. The study found that the highest percentages of cis-acting elements were related to light response, accounting for 31.09% of the total. Additionally, plant hormone-related cis-acting elements such as abscisic acid response elements (7), gibberellin response elements (11), salicylic acid response elements (5), methyl jasmonate response elements (30), and auxin response elements (1) were identified in promoter regions. There were 18 elements in the promoter region of *UrSINA1*, while there were only three elements in the promoter region of *UrSINA4*, indicating that *UrSINA1* might play an important role in plant growth and development, as well as in hormone response.

### 2.6. Ten UrSINA Genes Are Differentially Expressed in Various Tissues

In order to obtain the expression profiles of ten *UrSINA* genes in *U. rhynchophylla*, quantitative real-time polymerase chain reaction (qRT-PCR) was used to evaluate the expression levels of the UrSINA gene family in the roots, stems, and leaves of six-month-tissue-cultured seedlings of *U. rhynchophylla* (Figure 7). Ten *UrSINA* genes are expressed in the roots, stems, and leaves, with *UrSINA9* showing the highest expression levels in stems and leaves, while *UrSINA4* shows the lowest expression levels in leaves. All ten *UrSINA* genes appear to be expressed at similar levels in roots under normal conditions. And the expression levels of *UrSINA5/6/7* are indistinguishable among the roots and leaves. Furthermore, in synteny analysis, pairs of genes that exhibited collinearity also showed similar expression patterns (Figure 5b). For example, the expression levels of *UrSINA2* and *UrSINA*3 genes showed comparatively higher expressions in stems and lower expressions in roots.

### 2.7. Expression Analysis of UrSINAs and Key Enzyme Genes under ABA and MeJA Hormones

Previous studies have demonstrated that the *SINA* genes could respond to hormone responses [23,26]. To investigate whether *UrSINAs* also play a role in plant hormone responses, we evaluated their expression patterns in *U. rhynchophylla* treated with ABA and MeJA at different time periods. The qRT-PCR results show that the expression levels of *UrSINA3*, *UrSINA4*, *UrSINA7*, and *UrSINA10* exhibited no differential temporal expression patterns under ABA treatment, implying that they did not respond to the ABA hormone. Among them were *UrSINA4*, *UrSINA7*, and *UrSINA10*, which do not have ABA-responsive cis-acting elements (Figure 6). *UrSINA1*, *UrSINA2*, *UrSINA6*, and UrSINA9 showed differential temporal expression patterns. And *UrSINA9* was first up-regulated at 1 h and had the highest expression levels at 12 h after ABA treatment, indicating that *UrSINA9* is the most sensitive gene in response to ABA hormone (Figure 8). The same situation also occurs in kiwifruit and *Salvia miltiorrhiza*; for example, *AcSINA4/8/10/11* did not contain an ABRE cis-acting element in the promoter region, but they were all induced by ABA. No ABA responsive elements were found in *SmCUL2*, while it responded to ABA hormones [11,23].

Twelve key enzyme genes were also analyzed by qRT-PCR under ABA treatment. Here, 10 of 12 key enzyme genes showed a downward trend within 1–4 h, followed by an up-regulated trend thereafter. For example, *UrGES*, *UrTDC*, and *UrSGD* had the highest expression levels after 12 h of treatment, while *UrG10H*, *UrIO*, and *UrLAMT* had the highest expression levels after 24 h, and *Ur7DLGT*, *Ur7DLH*, *UrAS*, and *UrSTR* reached their peak expression levels at 48 h. On the contrary, *UrSLS* reached its peak expression level after 1 h of treatment and then showed a gradually decreasing expression trend. Then, co-expression correlation heatmaps between the 10 *UrSINA* genes and 12 key enzyme genes in the TIA biosynthesis pathway were constructed. *UrSINA1* was significantly positively (*r* > 0.8, *p* < 0.01) correlated with two key enzyme genes, *Ur7DLH* and *UrSTR*, respectively, while *UrSINA10* showed a significant negative correlation (*r* < −0.8, *p* < 0.01) with the expression pattern of *UrSLS*.

In general, the expression levels of 10 *UrSINA* genes exhibited a pattern of an initial increase, followed by a sharp decrease, and then a gradual rise after MeJA treatment. The 8 h treatment time point marked a significant shift in the responses of *UrSINA* genes to the MeJA hormone. Specifically, *SINA1, 2, 3, 9,* and *10* reached their peak gene expression levels after 12 h of treatment, showing a substantial increase compared to the control. Unlike the other *UrSINA* genes, *UrSINA4* displayed a relatively stable temporal expression pattern responding to MeJA, and this pattern is consistent with the observation that only *UrSINA4* lacks MeJA-responsive cis-acting elements (Figure 9a). The overall analysis indicates that *UrSINA* genes responded strongly to MeJA and had similar expression patterns, suggesting active responsiveness to this hormone. The expression patterns of key enzyme genes have been reported previously (Figure 9b) [4]. Co-expression correlation heatmaps between the 10 *UrSINA* genes and 12 key enzyme genes in the TIA biosynthesis pathway were also constructed. Only *UrSINA5* and *UrSINA6* were significantly positively (*r* > 0.8, *p* < 0.01) correlated with *UrAS* (Figure 9c).

## 3. Discussion

SINA E3 ubiquitin ligases are a family of RING E3 ubiquitin ligases and have been reported to be involved in various processes in plants, such as plant growth and development, hormone response, and biotic and abiotic stress [36,37]. The quantities of SINA genes, which signify the development and expansion of this gene family, vary among different plant species. Currently, there are five *AtSINA* genes (*AtSINAT1*-*AtSINAT5*), six *OsSINA* genes (*OsSINA1*-*OsSINA6*), six *ZmSINA* genes (*ZmSINA1*-*ZmSINA6*), six *MtSINA* genes (*MtSINA1*-*MtSINA6*), six *LjSINA* genes (*LjSINA1*-*LjSINA6*), six *SlSINA* genes (*SlSINA1*-*SlSINA6*), eleven *MdSINA* genes (*MdSINA1*-*MdSINA11*), and only one *MaSINA* gene identified from *A. thaliana*, *O. sativa*, *Zea mays*, *Medicago truncatula*, *Lotus japonicus, Solanum lycopersicum*, *Malus domestica*, and *Musa acuminata* [22,23,27,36,38,39]. In this work, 10 *UrSINA* genes were identified from the genome, and the number of *UrSINA* paralogous genes indicates a diversification of UrSINA functions in both spatial and temporal expression, as well as substrate specificity.

It has previously been found that some SINAs are located in the nucleus or cytoplasm [24]. However, according to the predicted results, the UrSINA proteins might be localized in the nucleus or mitochondria, indicating that the UrSINA protein might have different functions (Table 1). The results of multiple sequence alignment show that each UrSINA protein had a typical RING and SINA domain, with some differences at the N- and C-terminals (Figure 3). Moreover, a phylogenetic tree was constructed for the A. thaliana, O. sativa, and U. rhynchophylla proteins; 10 UrSINA proteins were unevenly categorized into two groups, with 3 proteins in Group I and 7 proteins in Group II (Figure 1). Intron diversity is critical for RNA stability, the regulation of gene expression, and alternative splicing; therefore, the intron has been considered a crucial characteristic of genes in relation to evolution [40]. The gene structures of UrSINA genes were analyzed in U. rhynchophylla. Almost all UrSINA genes contained the same gene structure, with three exons and two introns, except UrSINA1 and UrSINA7 (Figure 4). We also found that *UrSINA* genes formed collinear gene pairs, such as *UrSINA2*/*UrSINA3* and *UrSINA8*/*UrSINA9*, which had the same lengths of exons and introns (Figure 4c and Figure 5b). Similar results were found in many species, such as Arabidopsis, rice, and kiwifruit [22,23]. Therefore, it is hypothesized that these members of the SINA family are highly conserved throughout evolution and play a crucial role in *U. rhynchophylla*.

Gene duplication has had a significant impact on genome evolution, and tandem duplication, segmental duplication, or whole genome duplication have been regarded as dominant factors contributing to gene family expansion [41]. Collinearity analysis has shown that 5 of 10 UrSINA genes shared gene pairs, which was the main reason for the UrSINA gene family expansion. In addition, four (UrSINA2, UrSINA3, UrSINA8, and UrSINA9) of them simultaneously formed collinear gene pairs with AtSINA in Arabidopsis (Figure 5b,c; Appendix A). The comparison between the quantity of SINAT genes in Arabidopsis and the number of UrSINA genes in *U. rhynchophylla* implies gene expansion within this gene family over the course of evolution. The presence of four pairs of UrSINA genes suggests a duplication event in *U. rhynchophylla*, which might offer functional redundancy to this gene family.

Cis-acting elements are non-coding DNA sequences that play an important role in regulating gene transcription initiation and participate in various biological processes that regulate gene activity transcription in dynamic networks, including abiotic stress responses, hormones, and developmental processes [42,43]. A total of 119 cis-acting elements from 17 categories were obtained from the *UrSINAs* promoter region. Compared to the 13 types of 244 cis-acting elements obtained from kiwifruit, *UrSINA*s had a richer variety of cis-acting elements, with approximately one-third of them being light-responsive elements. Interestingly, it has been demonstrated that the degradation of SINATs is self-regulated in *A. thaliana*, and SINATs were degraded as a result of their interaction with the photoreceptors cryptochrome (CRY1) and phytochrome B (phyB) in the cytoplasm under dark conditions. Moreover, phyB and CRY1 were dissociated from SINATs by induced red and blue light, and this was the primary factor influencing the SINATs accumulation [44]. *UrSINA1* had the highest number and most diverse types of cis-acting elements, while *UrSINA4* only had three cis-acting elements. It is interesting that under the treatment of ABA and MeJA, the expression levels of *UrSINA1* were highly correlated with key enzyme genes on the pathway, especially under ABA treatment. Co-expression association analysis showed that *UrSINA1* was significantly positively correlated with the key enzyme genes *Ur7DLH* and *UrSTR* on the pathway, which may be related to the abundant cis-acting elements of *UrSINAs*.

Various *SINA* genes have been identified as key regulators of plant stress resistance in response to plant hormone signals. For instance, in *Arabidopsis*, SINAT2 works in conjunction with the receptor DSK2 to degrade BES1 during drought stress, thereby positively influencing the ethylene biosynthesis triggered by brassinosteroids [45]. Similarly, MdSINA2 in apple has been shown to negatively modulate the response to ABA, with overexpression in *Arabidopsis* leading to heightened sensitivity to abscisic acid [24]. In rice, the homologue of *Arabidopsis* SINAT5, OsDIS1, functions as a negative regulator in drought response, as evidenced by enhanced early resistance in transgenic rice resulting from silencing OsDIS1 [46]. According to the phylogenetic tree, UrSINA7 was clustered with OsSINA4 (OsDIS1) and OsSINA6 (Figure 2), implying that UrSINA7 may act as a negative regulatory factor in response to drought. Not only that, but previous studies have also shown that the expression of resistant genes were regulated by the MYB transcription factor. For instance, PbrMYB21 was able to positively contribute to drought tolerance by binding to the MYB recognition sites in the *PbrADC* promoter region [47]. Many MYB binding sites were identified in the promoter region of *UrSINA* genes (Figure 6); therefore, *UrSINA* genes might likely be regulated by the MYB transcription factors, which are involved in mediating drought-stress signaling. Additionally, in tomato, SlSINA3 modulates defense signals within the immune system by regulating the degradation of SINAC1. Notably, the overexpression of SlSINA2 in tomato impedes growth and induces leaf chlorosis, while the overexpression of SlSINA5 influences normal flower development [27]. The analysis of gene expression patterns is a critical method for studying gene function [7]. To explore the potential functions of *UrSINA* genes in *U. rhynchophylla*, we examined their expression patterns under normal conditions and various hormone treatments. It was observed that, under normal conditions, *UrSINA* genes could be expressed, suggesting that this gene family may play fundamental roles in *U. rhynchophylla*. Most *UrSINAs* exhibited higher expression levels in stems and relatively lower levels in roots (Figure 7). This is consistent with the expression patterns of the AcSINA genes in different varieties of kiwifruit [23]. Previous studies have shown that ABA could increase the content of TIAs by activating the expression of genes in the TIA biosynthesis pathway in *C. roses* [48]. We found that ABA can accumulate rhynchophylline and isorhynchophylline contents (data is unpublished). Most key enzyme genes are up-regulated under ABA treatment (Figure 8b). And according to co-expression, *UrSINA1* was significantly positively correlated with *Ur7DLH* and *UrSTR*, and UrSINA10 showed a negative correlation with *UrSLS* (Figure 8c). This result shows that the *UrSINA* gene and key enzyme genes could share similar regulatory effects as a result of the ABA treatments, implying that these *UrSINA* genes might participate in TIA biosynthesis in *U. rhynchophylla*.

## 4. Materials and Methods

### 4.1. Plant Material and Hormone Treatment

The materials used in the experiment were *U. rhynchophylla* tissue-culture seedlings from the Horticulture College of Hunan Agricultural University, identified by Tang Qi experts. The plants were grown in one-half MS medium for 6 months, and the medium was composed as follows: 3-indolebutyric acid, 0.2 mg/L; 1-naphthlcetic acid, 0.2 mg/L; sucrose, 30 g/L; agar, 8 g/L; and activated carbon, 0.2 g/L. Healthy plants at the same stage of development were grown in a greenhouse (25 °C ± 2) under a 12 h light/12 h dark photoperiod; the leaves were sprayed with a sterile solution of MeJA (Colaber, Beijing, China) at a concentration of 100 μM, the roots of the *U. rhynchophylla* seedlings were treated with a liquid medium containing 100 μM ABA, and the caps were fastened to prevent hormone evaporation into the air. Samples were collected at 1, 4, 8, 12, 24, and 48 h, with 0 h as the control. In order to detect the gene expression of *UrSINAs* in different parts of *U. rhynchophylla* tissue-culture seedlings, roots, stems, and leaves of 6-month-old tissue-culture seedlings were selected. For each time point, 3 plants were used as biorepeats. The whole-plant samples collected were immediately frozen with liquid nitrogen and stored at −80 °C for relative gene expression analysis.

### 4.2. Total RNA Extraction and Relative Gene Expression Analyses

Total RNA was extracted using the *SteadyPure* Plant RNA Extraction Kit (Accurate Biology, Changsha, China), and cDNA synthesis was performed using an *Evo M-MLV* RT Mix Kit with gDNA Clean (Accurate Biology, Changsha, China) to remove genomic DNA according to the manufacturer’s instructions. Quantitative real-time PCR was performed using the ABI 7300 (ABI, Alexandria, VA, USA). RT-qPCR procedures and methods were the same as those in previous reports [7]. The relative expressions of genes were calculated using the 2^−∆∆CT^ method with *SAM* as an internal reference gene. Three technical replications were used in each RT-qPCR reaction, and three biological replications were performed. RT-qPCR primers were designed using Beacon Designer 7.0 software (Appendix A). The GraphPad Prism 9 software was used to analyze data correlation and significant differences.

### 4.3. Identification of SINA Genes in U. rhynchophylla and Construction of Phylogenetic Tree

Candidate SINA members were identified by a hidden Markov model (HMM, PF031450) by in silico protein domain homology searches. The websites NCBI-CDD (https://www.ncbi.nlm.nih.gov/Structure/bwrpsb/Bwrpsb.cgi, accessed on 12 May 2024), SMART (http://smart.embl.de/, accessed on 12 May 2024), and ExPASy (https://prosite.expasy.org/, accessed on 12 May 2024) were used to characterize the integrity of SINA structural domains. In addition, the website ExPASy (https://web.expasy.org/protparam/, accessed on 14 May 2024) was utilized to analyze the physical and chemical properties of UrSINA proteins. Finally, subcellular localization prediction was performed using the website Plant-PLoc 2.0 (http://www.csbio.sjtu.edu.cn/bioinf/plant-multi/, accessed on 15 May 2024). From the Arabidopsis information database TAIR (https://www.arabidopsis.org/index.jsp, accessed on 15 May 2024) and the rice genome database (http://rice.plantbiology.msu.edu/cgi-bin/ORF-infopage.cgi, accessed on 15 May 2024), the corresponding SINA protein sequences were downloaded, respectively. MEGA 7.0 software was used to construct the phylogenetic tree of SINA protein sequences of *U. rhynchophylla*, *A. thaliana*, and *O. sativa* [49].

### 4.4. Multiple Sequence Alignment, Conserved Motif, and Gene Structure Analysis

The SINA protein sequences of *U. rhynchophylla* were aligned by multi-sequence alignment using DNAMAN 7.0 software. Conserved motif analysis was performed using the software MEME (http://meme-suite.org/tools/meme, accessed on 17 May 2024), with the following conditions: maximum number of motifs = 10 and other parameters as default [50]. The gene structure information of the UrSINA family members was obtained from the GFF annotation file of the *U. rhynchophylla* genome. The gene structure was visualized and beautified using the (VisualizeGenestructure{fromGTF/GFF3File}) function of the TBtools software v2.096 [33].

### 4.5. Chromosome Localization, Syntenic Analysis, and Cis-Regulatory Elements Analysis

The distribution of UrSINA family members on chromosomes was analyzed using TBtools (gene location visualize from GTF/GFF) software, named according to the order of *UrSINA* distribution on the chromosome. To investigate the evolutionary relationship of *UrSINA*, a collinearity analysis of *SINA* genes’ intraspecific and interspecific species (*A. thaliana*, *O. sativa*, and coffee) was performed using the Multiple Collinearity Scan toolkit (MCScanX) of TBtools [34]. The 2000 bp promoter region upstream of the *UrSINA* gene was extracted using TBtools software, and the online website PlantCARE (http://bioinformatics.psb.ugent.be/webtools/plantcare/html/, accessed on 19 May 2024) was used to predict the cis-acting elements of all *UrSINA* genes [51].

## 5. Conclusions

In summary, we identified the SINA gene family in *U. rhynchophylla* based on genomic data and qRT-PCR expression analyses. Ten UrSINA proteins with complete structural domains were identified, and their physicochemical properties, gene structure, conserved motifs, sequence characterization, cis-acting elements in the promoter region, collinear analysis, and expression patterns were analyzed to explore the possible role of UrSINA in *U. rhynchophylla*. Most UrSINA genes were abundantly expressed in the stems and leaves of *U. rhynchophylla*. The expression levels of *UrSINA* and key enzyme genes in the pathway were detected by qRT-PCR with ABA and MeJA treatment, and it was found that *SINA1, 2, 5, 6*, and *9* might respond to ABA and MeJA stresses. The results of co-expression association analysis show that *UrSINA1* may be involved in the biosynthesis of TIAs, which provides a theoretical basis for understanding the biological functions of the SINA gene in plant secondary metabolites and growth and development.

## Figures and Tables

**Figure 1 ijms-25-07636-f001:**
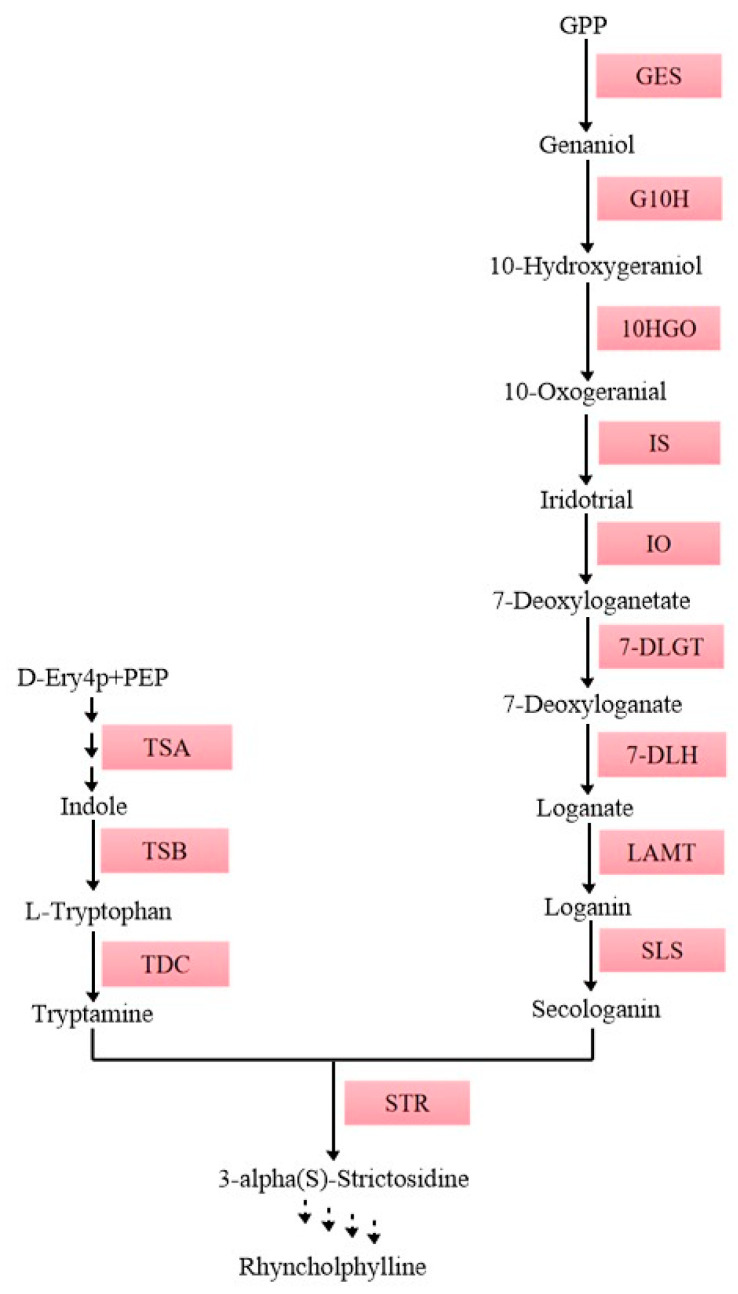
The biosynthetic pathway for Rhynchophylline and Isorhynchophylline in *U. rhynchophylla.* Solid arrows represent identified genes of this pathway, and dashed arrows represent presumed genes of the pathway. GES, geraniol synthase; G10H, geraniol 10-hydroxylase; 10-HGO, 10-hydroxy-geraniol oxidoreductase; IS, iridoid synthase; IO, iridoid oxidase; 7-DLGT, 7-deoxyloganetic acid glucosyltransferase; 7-DLH, 7-deoxyloganic acid hydroxylase; LAMT, loganic acid O-methyltransferase; SLS, secologanin synthetase; STR, strictosidine synthase; TSA, tryptophan synthase alpha; TSB, tryptophan synthase beta; TDC, tryptophan Decarboxylase.

**Figure 2 ijms-25-07636-f002:**
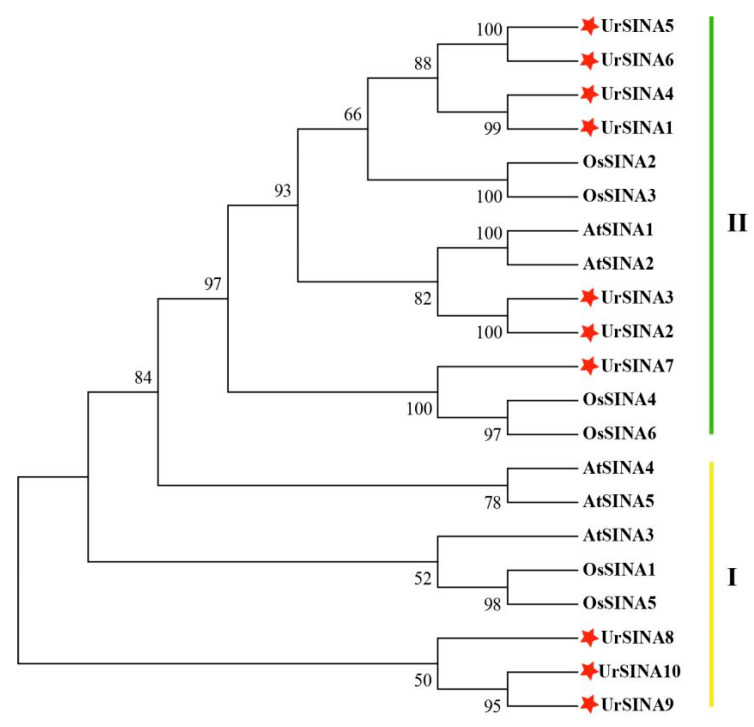
The phylogenetic tree of SINA proteins from *A. thalina*, *O. sativa*, and *U. rhynchophylla* constructed using MEGA 7.0 software with the neighbor-joining method containing 1000 bootstrap replicates. The red stars represent UrSINA proteins.

**Figure 3 ijms-25-07636-f003:**
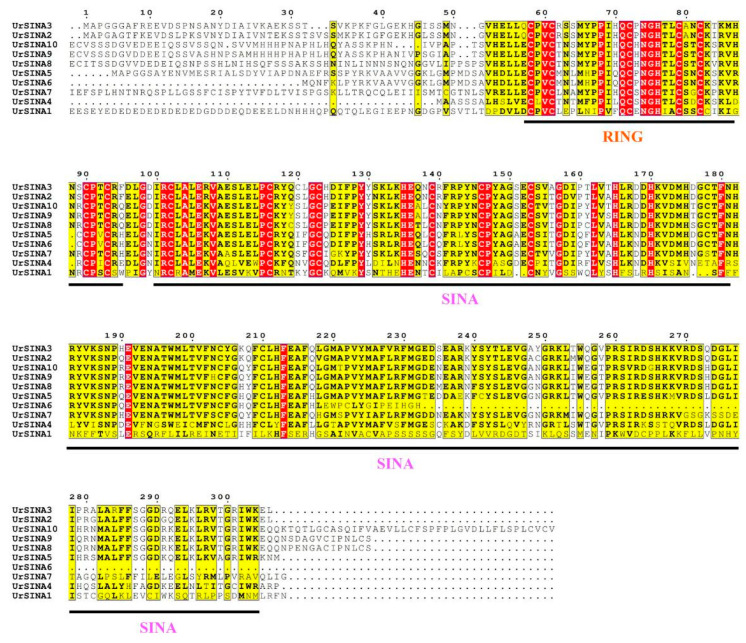
The multiple sequence alignment of ten UrSINA proteins using ESPript 3.0 online website. The most conservative domains are marked in red.

**Figure 4 ijms-25-07636-f004:**
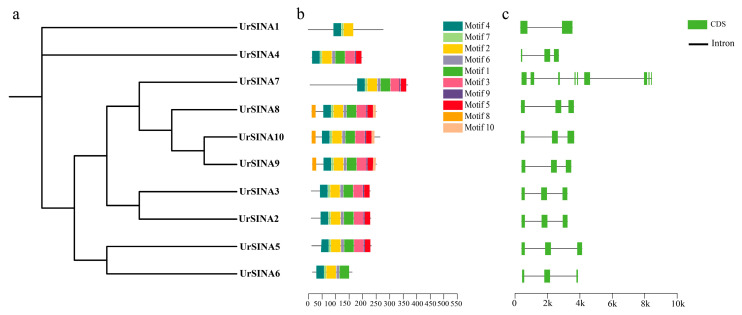
Conserved motifs and gene structure of SINA gene family in *U. rhynchophylla*. (**a**). Phylogenetic tree of 10 UrSINA proteins. (**b**). The conserved motif analysis of SINA proteins in *U. rhynchophylla*. Different motifs are colored with different colors. (**c**). Exon and intron structures of *UrSINA* genes.

**Figure 5 ijms-25-07636-f005:**
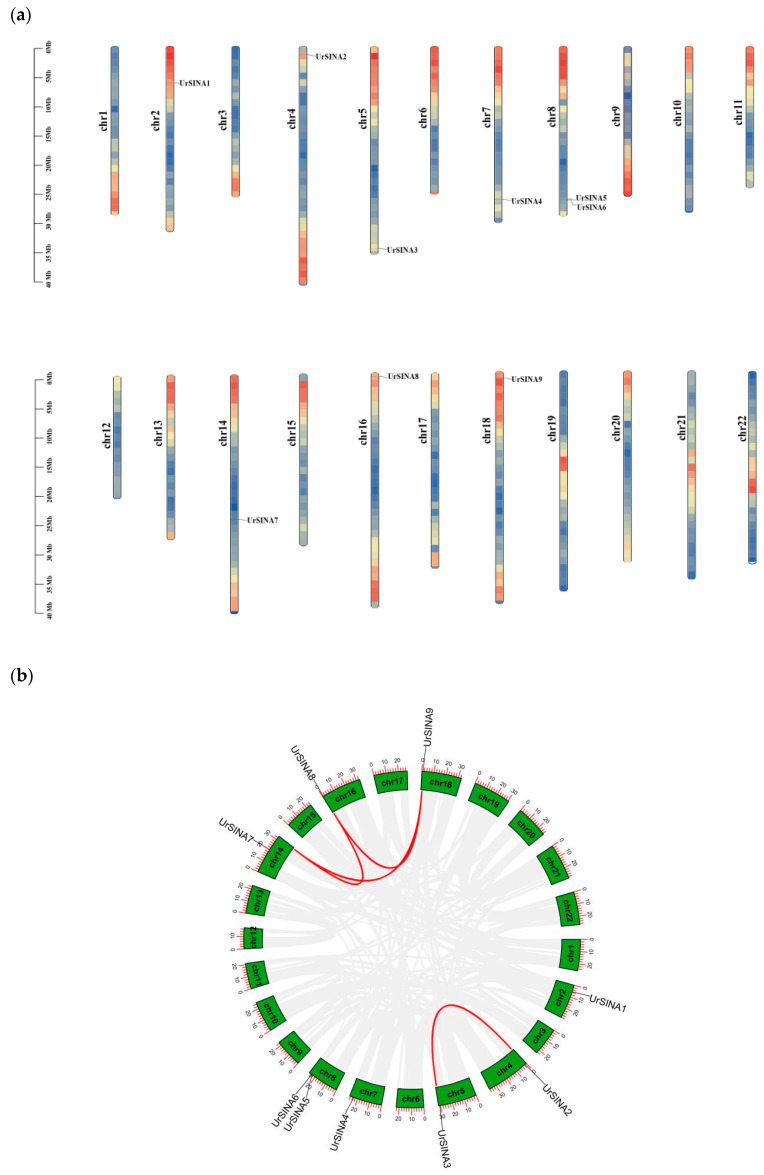
Chromosomal distribution and synteny analysis of *UrSINA* genes. (**a**). *UrSINA* genes on 22 chromosomes of *U. rhynchophylla*. The scale bar on the left indicates the length of chromosomes (Mb). Red represents high chromosome density, while blue represents low chromosome density. (**b**). The interspecific collinearity analysis of *UrSINAs*. The circle plot was created with MCScanX. Identified collinear genes are linked by red lines. (**c**). Syntenic relationship of *SINA* genes among *U. rhynchophylla*, *A. thaliana*, *O. sativa*, and *C. canephora*. Collinear *UrSINAs* identified are marked with red lines.

**Figure 6 ijms-25-07636-f006:**
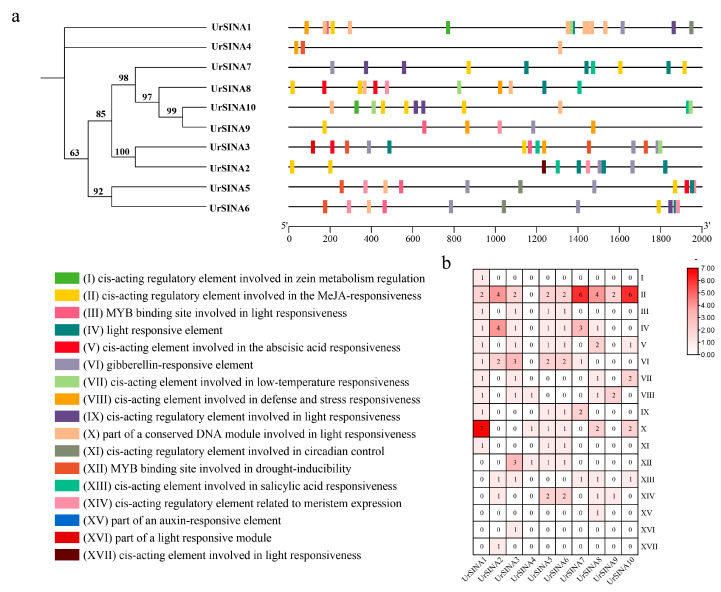
The analysis of cis-acting elements in the promoter region of the *UrSINA* genes. (**a**). Predicted cis-acting elements. (**b**). Number of cis-acting elements. The darker the color, the more elements there are. Different cis-acting elements are represented by rectangles of different colors.

**Figure 7 ijms-25-07636-f007:**
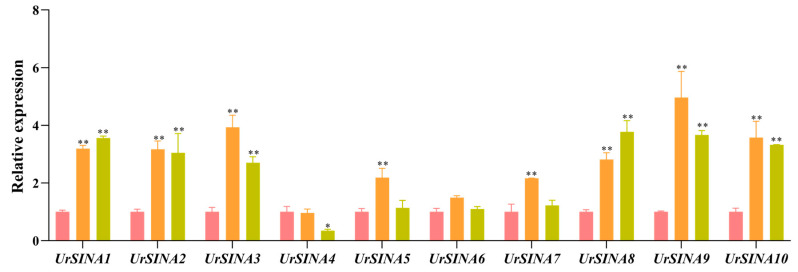
Expression patterns of ten *UrSINA* genes in different tissues. Roots, stems, and leaves were collected from six-month-tissue-cultured seedlings of *U. rhynchophylla*. *UrSAM* was used as the reference gene in this experiment. Values are means and standard deviations of three biological replicates. * and ** indicate *p*-values < 0.05 and <0.01, respectively. The color pink, orange, and yellow represent roots, stems, and leaves.

**Figure 8 ijms-25-07636-f008:**
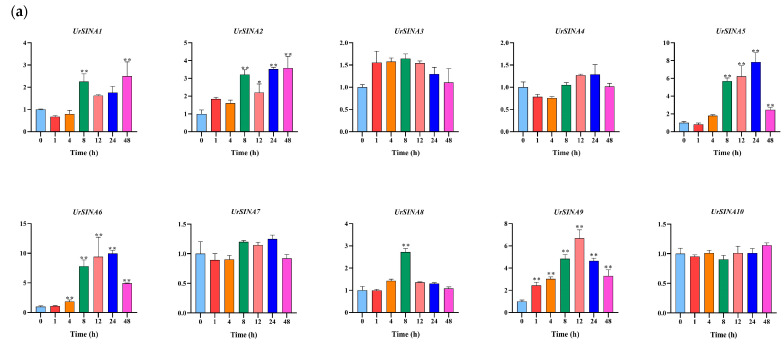
Expression patterns of the key enzyme genes and *UrSINA* genes in roots under ABA treatment. (**a**). Expression analysis of 10 *UrSINA* genes in *U. rhynchophyll*a roots under ABA treatment. (**b**). Expression patterns of the 12 key enzyme genes in roots under ABA treatment. (**c**). The correlation between the gene expression patterns of *UrSINA* and related genes in the pathway in roots under ABA treatment. The red represents positively correlated, and blue represents negatively correlated. The final results were calculated as the mean ± standard deviation. * and ** indicate *p*-values < 0.05 and <0.01, respectively.

**Figure 9 ijms-25-07636-f009:**
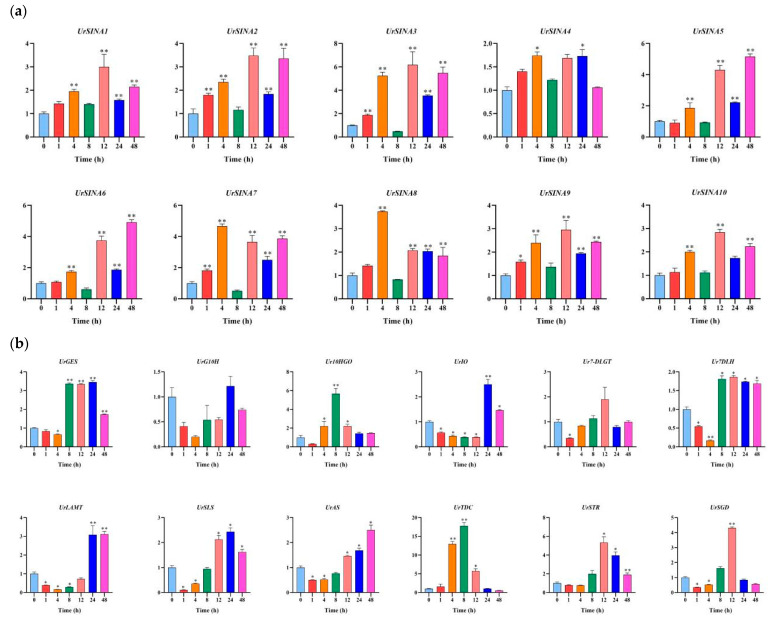
Expression patterns of the key enzyme genes and *UrSINA* genes in leaves under MeJA treatment. (**a**). Expression analysis of 10 UrSINA genes in U. rhynchophylla leaves under MeJA treatment. (**b**). Expression patterns of the 12 key enzyme genes in leaves under MeJA treatment. (**c**). Correlation heatmaps between the gene expression patterns of UrSINA and related genes in the pathway in leaves under MeJA treatment were constructed using TBtools. The red represents positively correlated, while white represents negatively correlated. The final results were calculated as mean ± standard deviation. * and ** indicate *p*-values < 0.05 and <0.01, respectively.

**Table 1 ijms-25-07636-t001:** The detailed information of ten *UrSINA* genes of *U. rhynchophylla.*

Gene ID	Gene Name	Chr	Start	End	Amino Acids (aa)	Molecular Weight (kDa)	Theoretical pI	Instability Index	Subcelluar Localization
g6840.t1	*UrSINA1*	chr2	5552559	5556024	388	44,051.84	4.51	74.28	Nucleus.
g29173.t1	*UrSINA2*	chr4	1326540	1329600	309	34,835.01	8.39	45.37	Mitochondrion. Nucleus.
g21951.t1	*UrSINA3*	chr5	30647250	30650297	306	34,688.69	8.5	45.78	Mitochondrion. Nucleus.
g29806.t1	*UrSINA4*	chr7	23211096	23213614	262	29,574.07	6.73	38.09	Nucleus.
g26498.t1	*UrSINA5*	chr8	23132312	23136340	308	34,923.39	7.84	48.27	Mitochondrion. Nucleus.
g41916.t1	*UrSINA6*	chr8	23289827	23293540	206	23,637.45	6.92	40.87	Mitochondrion. Nucleus.
g27550.t1	*UrSINA7*	chr14	20226758	20235432	506	57,458.84	6.82	37.63	Mitochondrion. Nucleus.
g4133.t1	*UrSINA8*	chr16	514255	517777	334	37,684.73	6.37	52.39	Mitochondrion. Nucleus.
g41269.t1	*UrSINA9*	chr18	927300	930607	332	37,683.68	6.53	48.82	Mitochondrion. Nucleus.
g26576.t1	*UrSINA10*	Scaffold9	387231	390770	353	40,128.82	6.46	51.41	Mitochondrion. Nucleus.

## Data Availability

The datasets generated and analyzed during the current study are available in NCBI. (GES:OP669346, G8H:OP669347, 8-HGO:OP669348, 7-DLS:OP669349, 7-DLGT:OP669350, 7-DLH:OP669351, SLS:OP669352, SGD:OP669353, STR:OP669354, AS:OP669355, AnPRT:OP669356, PRAI:OP669357, TSA:OP669358, TDC:OP669359.)

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
