# Peer review of "Genome-Wide Identification of Seven in Absentia E3 Ubiquitin Ligase Gene Family and Expression Profiles in Response to Different Hormones in Uncaria rhynchophylla"

_ijms, 2024, doi:10.3390/ijms25147636_

Round 1
Reviewer 1 Report
Comments and Suggestions for Authors
The paper describes the SINA E3 ubiquitin ligases family and the expression pattern of the SINA gene in Uncaria rhychophylla using RT-qPCR. The expression patterns of SINA genes were compared with those of genes of the terpenoid indole alkaloids pathway in a correlation analysis. I think the paper is of interest to science. The paper should benefit from the attention of the following comments.
Line 37-41, information on the present pathway is needed. Where is the pathay coming from? Is there a reference reported to the figure? Information on each enzyme in the figure is necessary.
Line 44-45, I think ABA is not completely an abiotic factor. Passage needs precision on that.
Line 53-54, please specify how the influence of secondary metabolism on the regulation of stress-responsive genes could be.
Line 112, precisions are required for the term 'the integrity of the conserved domain'. What information is incorporated into this activity?
Reference 18, please provide a precise reference for the passage.
Line 86-87, reference is required
Does SlSINT5 or sisia5?
Line 91-92, is a Sina ring
119-120, the stability adscription is not clear.
Figure 1, I think the emphasis on the enantiomeric compound is not necessary since the paper does not generate data on it.
Figure 3, is not mentioned in the results section.
Figure 4, lacks panels as figure legends
Figure 5, panel a, does not make sense to include chromosomes without UrSINAs.
Lines 166-167, do not make sense since only two genes map to the same chromosomes. It is evident that tandem duplication of results is not something that should be expected.
Figure 6b is repetitive information, which is actually depicted in figure 6a. Also, the scale in figure 6b does not include the entire range of values.
Line 218-219, Double check the passage, it seems that UrSINA4 is the lowest expressed gene.
Line 246, Double check the passage. It seems that it is related to enzymes that were analyzed.
325 327, I think the conclusion should be weakened as some SINA genes are specific to species rather than extensible across species.
329 332, I think the two passages contradict each other, and the conclusion should focus on the main findings.
Lines 338-341 and 347-349. Where is the data coming from? What is the link between kiwifruit and the findings reported in this paper? Is it coffee instead of kiwifruit?
Lines 397-398, It is not clear why the whole transcriptome data was mentioned.
Figure 9 is not mentioned in the main text and its panel b is located on the left side.
Line 412, Does the seedling media fresmedia replace itself every 6 months?
Line 416, Specified the solvent for the stock and spray solution of the material
Line 436, why transcription factors?
Line 437, genome sources required
Line 430, The paper should benefit from adding the minimal requirements for qPCR repeatability required (such as the MIQE guidelines)
Table 1, GenBank ID should be useful. Units in the columns should be beneficial for the readability of the table.
Fix any typos in the paper, for instance line 498 and 317
Author Response
Please see the attachment. Thank you !

Reviewer 2 Report
Comments and Suggestions for Authors
The English use and grammar made by a professional language editor are highly recommended to avoid syntax errors.
Round 2
Reviewer 1 Report
Comments and Suggestions for Authors
The comments have been addressed. I don't have additional comments.
Reviewer 2 Report
Comments and Suggestions for Authors
I am satisfied with the current revised version.